# Investigation of Genes and Proteins Expression Associating Serotonin Signaling Pathway in Lung and Pulmonary Artery Tissues of Dogs with Pulmonary Hypertension Secondary to Degenerative Mitral Valve Disease: The Preliminary Study

**DOI:** 10.3390/vetsci9100530

**Published:** 2022-09-28

**Authors:** Nattawan Tangmahakul, Siriwan Sakarin, Somporn Techangamsuwan, Anudep Rungsipipat, Sirilak Disatian Surachetpong

**Affiliations:** 1Department of Veterinary Medicine, Faculty of Veterinary Science, Chulalongkorn University, Bangkok 10330, Thailand; 2Companion Animal Center Research Unit (CAC-RU), Department of Pathology, Faculty of Veterinary Science, Chulalongkorn University, Bangkok 10330, Thailand

**Keywords:** canine, degenerative mitral valve disease, gene, lung, protein, pulmonary artery, pulmonary hypertension, serotonin

## Abstract

**Simple Summary:**

Pulmonary hypertension is an unusual elevation of the blood pressure in pulmonary circulation. In dogs, pulmonary hypertension is commonly occurred as a complication of progressive degenerative mitral valve disease especially in senior small-breed dogs. Previous studies revealed that the serotonin signaling pathway is associated with the pathogenesis of pulmonary hypertension. However, research in dogs was scarce. Therefore, the present study aimed to illustrate the understanding of this point by assessing the expression of the targeted genes and proteins related to the serotonin pathway in lung tissues and pulmonary arteries of dogs. Our results showed that the pattern of gene and protein expression was different in canine lung and pulmonary arteries. The targeted proteins in pulmonary artery tissues of the degenerative mitral valve disease dogs with and without pulmonary hypertension tended to be upregulated. In addition, the expression of these protein was increased in the pneumocytes of the alveolar walls, pulmonary alveolar macrophages, and bronchial epithelial cells. A correlation between the targeted gene and protein expression and the echocardiographic data was also founded. Overall results pointed out that the serotonin pathway in lung and pulmonary artery tissues might have different roles in degenerative mitral disease with and without pulmonary hypertension.

**Abstract:**

Pulmonary hypertension (PH) is defined as an increase in pulmonary vascular pressure. It is one of the most common complications that occur as a result of degenerative mitral valve disease (DMVD) in dogs. Serotonin (5-HT) can trigger the development of PH. Accordingly, this study investigated the changes in the expression of genes and proteins associated with local 5-HT signaling in the lungs and pulmonary arteries (PA) of dogs with PH secondary to DMVD. Lung and PA tissue samples were collected from the cadavers of fourteen small-breed dogs and divided into normal (*n* = 4), DMVD (*n* = 5) and DMVD with PH (*n* = 5) groups. Gene expression (*tph1*, *slc6a4* and *htr2a*) was analyzed by quantitative reverse transcription-polymerase chain reaction (qRT-PCR). The expression of proteins (TPH-1, SERT, 5-HTR2A, ERK1/2 and pERK1/2) was examined by Western blot analysis and immunohistochemical staining. The results showed that the expression of genes and proteins evaluated by qRT-PCR and Western blot analysis in lung and PA tissues did not differ among groups. However, the expression of proteins related to 5-HT signaling tended to be upregulated in PA tissues from DMVD dogs with and without PH. Immunohistochemical examination revealed the overexpression of these proteins in the DMVD and DMVD with PH groups in lung tissue. These findings suggest a local effect of 5-HT signaling in DMVD dogs with and without PH.

## 1. Introduction

Pulmonary hypertension (PH) is characterized by increased mean pulmonary arterial pressure (mPAP) [1]. The American College of Veterinary Internal Medicine categorized canine PH into six categories [2]. The most common cause of PH in dogs is the PH due to left heart disease [2,3]. In dogs affected with degenerative mitral valve disease (DMVD), chronic mitral regurgitation increases left atrial pressure, leading to pulmonary vein congestion and pulmonary oedema. This is followed by pulmonary artery constriction and remodeling, leading to the development of PH [4,5]. The incidence of PH in dogs with DMVD ranged from 13.94−53.33% [6,7,8,9,10]. PH is common in more severe forms of DMVD. In addition, survival time is shorter in DMVD dogs with PH than those without PH [6].

Serotonin or 5-hydroxytryptamine (5-HT) is a substance in the central nervous system and a biogenic amine in the periphery that is abundant in the intestinal wall and platelets [11]. In the peripheral pool, 5-HT is synthesized mainly in enterochromaffin cells. Tryptophan hydroxylase-1 (TPH-1) is an enzyme that converts tryptophan to 5-HT [12,13]. After synthesis and release into the bloodstream, 5-HT is kept in the platelets by uptake via 5-HT transporters (SERT) and receptors [14,15,16]. It is metabolized to 5-hydroxyindoleacetic acid (5-HIAA) and excreted in the urine [17,18]. Assessment of 5-HIAA level reflects the cellular serotonin degradation, however, 5-HIAA did not stimulate the proliferation of pulmonary artery smooth muscle cells (PASMCs) [19]. It was found that 5-HT plays an important role in the physiology and pathology of various organs [11,20,21,22,23]. Moreover, 5-HT was hypothesized that it can cause PH because 5-HT levels have found increase in women treated with SERT substrate anorectic drugs, resulting in an increased risk of PH [24,25]. In lung tissue, pulmonary artery endothelial cells (PAECs) produces 5-HT and transmitted in a paracrine manner to PASMCs via SERT and 5-HT receptors [26]. It then stimulates downstream effector pathways, which are Rho/Rho kinase, and mitogen-activated protein kinase kinase/extracellular signal-related kinase pathways [27,28,29]. These pathways induce the phosphorylation of ERK1/2 and translocation from cytoplasm to nucleus of PASMC. In PASMC nucleus, phosphorylated ERK1/2 (pERK1/2) causing pulmonary artery vasoconstriction and remodeling [27,28,29,30]. In humans and animal models, there is evidence for the linking of 5-HT signaling to the development of PH secondary to several causes [31,32]. In addition, the expression of proteins related to 5-HT signaling in the pulmonary artery (PA) of PH dogs due to DMVD has been reported [33]. However, the expression of genes and proteins related to 5-HT signaling in the lung tissue of PH dogs due to DMVD has not been studied. The objective of this study was to evaluate the expression of genes and proteins reflecting the local signaling of the 5-HT pathway in lung and PA tissues of DMVD dogs with PH compared with DMVD dogs without PH and normal dogs.

## 2. Materials and Methods

### 2.1. Animals

The cadavers of small-breed dogs aged 7–15 years, with a body weight of <10 kg, were recruited. Necropsy of all enrolled dogs was performed at the Department of Pathology, Faculty of Veterinary Science, Chulalongkorn University, Thailand. The history and clinical evaluation were used to enroll and classify the dogs into three groups, including normal, DMVD, and PH due to DMVD (DMVD with PH) groups. The normal group consisted of dogs that died naturally without systemic diseases and abnormalities in the cardiorespiratory system. Dogs were presented with thin and translucent mitral valve leaflets or thickness less than 2 mm measured with calipers at necropsy. DVMD dogs with stage C and D without PH were included in the DMVD group. Dogs with DMVD stage C or D were dogs with valve thickening, prolapse, and regurgitation with left-sided heart enlargement (Figure 1) and pulmonary edema [34]. The DMVD with PH group included DMVD stage C or D dogs with an intermediate or high probability of PH. Dogs that had developed PH from other causes, such as respiratory disease and heartworm infestation, were excluded. The peak velocity of tricuspid regurgitation (TR) (Figure 1) and the number of anatomic sites with echocardiographic evidence of PH were used to evaluate PH probability following the ACVIM consensus guidelines [2]. Dogs affected with DMVD with and without PH received drugs to control heart failure condition, including benazepril or ramipril, furosemides, and pimobandan. There was no dog that were prescribed with drugs that affected serotonin signaling pathway such as tramadol, ondansetron, metoclopramide, dextromethorphan, or other drugs [35]. At necropsy, the lungs and pulmonary arteries were dissected and collected. Tissue samples were stored at −80 °C for RT-PCR and Western blot Appendix A analysis and preserved in 10% formalin for 24 h for immunohistochemistry.

### 2.2. Investigation of Genes Related to 5-HT Signaling Pathway

The NucleoSpin RNA kit was used for the RNA extraction (Macherey-Nagel, Allentown, PA, USA). The TURBO DNA-*free*™ kit (Invitrogen, Watham, MA, USA) was applied on total RNA to remove genomic DNA (gDNA. The Omniscript RT kit (Qiagen, Hilden, Germany) was added to gDNA for reverse transcribing into complementary DNA (cDNA). The cDNA samples from all dogs were then subjected to relative quantitative qRT-PCR (Roter-Gene Q, Qiagen, Taufkirchen, Germany) using SYBR Green I dye (KAPA SYBR Green Fast qPCR kit Mastermix (2X) Universal, Kapabiosystems, Wilmington, MA, USA) and the specific primers for the target genes, *tph1*, *slc6a4*, *htr2a*, and the internal reference genes, *RPS19* for PA tissue [36] or *RPL32* for lung tissue [37] (Table 1). The online-program, Primer 3 Plus was used to design primers for the target genes, and the appropriate internal reference genes used to normalize the target genes were selected from the previous studies [36,37]. qRT-PCR of the target genes was performed in triplicate at 95 °C for 3 min, followed by 40 cycles of denaturation at 95 °C for 3 s, annealing at 63 °C for 25 s for *tph1* and *slc6a4* and 30 s for *htr2a*, and extension at 72 °C for 30 s. Reference genes were run in triplicate at 95 °C for 5 min, followed by 40 cycles of 95 °C for 5 s and 60 °C for 25 s. The melting curve of all reactions was analyzed at a rate increase of 1 °C per second at 65–95 °C. The comparative C_T_ method was used to analyze the relative quantitation of targeted gene expression [38].

### 2.3. Investigation of the Downstream Effectors in 5-HT Signaling Pathway by Western Blot Analysis

TPH-1, SERT, serotonin receptor 2A (5-HTR2A), ERK1/2 and pERK1/2 were assessed and normalized by the reference protein, α-tubulin [40]. Lung and PA tissues were ground under aseptic conditions with liquid nitrogen. The tissue powder was then homogenized by adding RIPA buffer (Pierce^®^ RIPA Lysis and Extraction Buffer, Thermo Fisher Scientific, Waltham, MA, USA), protease inhibitor (Halt™ Protease Inhibitor Cocktail, EDTA Free, Thermo Fisher Scientific, Waltham, MA, USA) and phosphatase inhibitor (Sodium orthovanadate, Sigma-Aldrich, Darmstadt, Germany). The tissue lysate was centrifuged at 15,000× *g* for 30 min at 4 °C and the supernatant was collected. The Bradford protein assay was conducted using Bio-Rad Protein Assay Dye Reagent Concentrate (Bio-Rad^®^, Hercules, CA, USA) and the protein concentration of the samples was measured at the wavelength of 595 nm by the spectrophotometer (Spectronic^®^ 20 GENESYS™, Thermo Fisher Scientific, Waltham, MA, USA). Samples were prepared by adding the sodium dodecyl sulfate (SDS) sample buffer (2× Laemmli Sample Buffer, Bio-Rad^®^, Hercules, CA, USA) and β-mercaptoethanol. Samples were heated at 95 °C for 5 min. Except for the assay of SERT, samples were heated at 85 °C for 2 min. Samples containing 50 µg of total protein were separated on a 10% SDS/polyacrylamide gel and transferred to a polyvinylidene difluoride (PVDF) membrane (Immun-blot PVDF membrane, Bio-Rad^®^, Hercules, CA, USA). Membranes were blocked with 2.5% BSA for 30 min for SERT, 1 h for 5-HTR2A and α-tubulin, and 5% BSA for 1 h for TPH-1, ERK1/2, and pERK1/2. The membranes were incubated overnight at 4 °C with 1:500 mouse anti-TPH-1 antibody (mouse monoclonal anti-tryptophan hydroxylase antibody, Sigma-Aldrich, Darmstadt, Germany), 1:1000 mouse anti-SERT antibody (mouse monoclonal anti-SERT antibody, Advanced Target Solutions, Kentwood, MI, USA), 1:1000 mouse anti-5-HTR2A antibody (mouse monoclonal anti-SR-2A antibody, Santa Cruz Biotechnology, Santa Cruz, CA, USA), 1:1000 rabbit anti-ERK1/2 antibody, 1:1000 rabbit anti-pERK1/2 antibody (rabbit monoclonal anti- Phospho Plus p44/42 MAPK (Erk1/2) (Thr202/Tyr204) antibody Duet, Cell Signaling Technology, Danvers, MA, USA), and 1:1000 mouse anti-α-tubulin antibody (mouse monoclonal anti-alpha tubulin antibody, Invitrogen™, Waltham, MA, USA). The membranes were then incubated with horseradish peroxidase-conjugated anti-rabbit/mouse secondary antibody (EnVision Detection Systems, Peroxidase/DAB, rabbit/mouse, Agilent Technologies, Santa Clara, CA, USA) and detected with 3,3′-diaminobenzidine (DAB). The measurement of protein band intensity was performed using ImageJ software (NIH, Besthesda, MA, USA).

### 2.4. Investigation of the Downstream Effectors in 5-HT Signaling Pathway by Immunohistochemistry

The 4-µm-thick lung sections were deparaffinized in xylene, rehydrated in graded alcohol, and then treated with citrate buffer (0.01 M, pH 6.0). Endogenous peroxidase activity was blocked with 3% hydrogen peroxide and 1% bovine serum albumin. Slides were applied with the primary antibodies overnight at 4 °C. The primary antibodies used for immunohistochemical staining were the same as those used for Western blot analysis. All antibodies were used at a dilution of 1:200. Then, horseradish peroxidase-conjugated anti-rabbit/mouse secondary antibodies (EnVision Detection Systems, Peroxidase/DAB, Rabbit/Mouse, Agilent Technologies, Santa Clara, CA, USA) were then applied and the color reaction was detected by incubation with 3,3′-diaminobenzidine (DAB) and counterstaining with Mayer’s hematoxylin. The photographs of each lung section were taken with a photomicroscope (Olympus BX50, Tokyo, Japan) at 40× magnification for 10 areas to evaluate the expression of TPH-1, SERT, 5-HTR2A, ERK1/2 and pERK1/2. The number of positively stained cells was counted manually and calculated as the percentage of positive cells relative to the total number of cells.

### 2.5. Statistical Analysis

Relative gene and protein expression evaluated by qRT-PCR and Western blot analysis was analyzed by the Kruskal–Wallis test and Dunn’s post-hoc test. The percentage of positive cells from immunohistochemical study was expressed as mean ± standard deviation (SD) and analyzed by one way ANOVA. The Bonferroni test was used for post-hoc analysis. The correlation between the relative expression of all targeted genes and proteins and echocardiographic data was analyzed by Spearman correlation. The computer-based program, SPSS version 22 (IBM, Armonk, NY, USA) was used for statistical analysis. *p*-value less than 0.05 was considered statistical significance.

## 3. Results

### 3.1. Animals

Fourteen canine cadavers were included and divided into normal (*n* = 4), DMVD (*n* = 5), and DMVD with PH (*n* = 5) groups. The signalment including gender, age, body weight and breed of the enrolled dogs, stages of DMVD and the medications that dogs received were shown in Table 2. Normal dogs were dead with the diseases or conditions which did not induce any cardiovascular disease and/or PH. Dogs in the DMVD and DMVD with PH groups died from the adverse progression of the disease including congestive heart failure. Echocardiographic data showed no difference between parameters in the DMVD and DMVD with PH groups, except for fractional shortening (FS), normalised left ventricular internal diameter at the end of diastole (LVIDdN), and normalised left ventricular internal diameters at the end of systole (LVIDsN). Echocardiographic data are shown in Table 3.

### 3.2. Investigation of Genes Associating 5-HT Signaling Pathway

This study showed that the expression of *tph1*, *slc6a4*, and *htr2a* was found in both lung and PA tissues. There was no difference in *tph1*, *slc6a4*, and *htr2a* expressions among the groups. The expression pattern of the *tph1*, *slc6a4*, and *htr2a* was different between lung and PA tissues (Table 4). In lung tissues, the relative *tph1* expression in PA tissues tended to be upregulated in the DMVD and DMVD with PH groups. The trend of downregulation of *slc6a4* was found in the lung tissue of the DMVD and DMVD with PH groups. However, this trend was not similar in PA tissue. The relative expression of *htr2a* in lung and PA tissues tended to be upregulated in the DMVD with PH group compared to the DMVD group.

### 3.3. Investigation of Proteins Associating 5-HT Signaling Pathway

#### 3.3.1. Western Blot Analysis

Expression of the proteins examined in this study was found in both lung and PA tissues. According to Western blot analysis, no difference in protein expressions among the groups, but a tendency for changes was noted. The pattern of protein expression was different in lung and PA tissues. In lung tissue, the expression patterns for TPH-1, SERT, 5-HTR2A, ERK1/2, and pERK1/2 proteins could not be determined, but TPH-1 and 5-HTR2A tended to be upregulated in the PA tissues of dogs in the DMVD and DMVD with PH groups. In addition, the expression of SERT and ERK1/2 tended to be upregulated in the DMVD with PH groups. The relative protein expression in the present study is concluded in the Table 5.

#### 3.3.2. Immunohistochemical Study

The TPH-1, SERT, 5-HTR2A, ERK1/2, and pERK1/2 were expressed by pneumocytes of the alveolar wall, pulmonary alveolar macrophages (PAMs), and bronchial epithelial cells. Expression of TPH-1, SERT and 5-HTR2A was observed in the cytoplasm, whereas downstream signaling proteins, including ERK1/2 and pERK1/2, were expressed in the nucleus and cytoplasm. Quantitative analysis in Table 6 showed that percentages of TPH-1-, SERT- and 5-HTR2A- positive pneumocytes were extremely low in the control group and increased in the DMVD and DMVD with PH groups (Figure 2 and Figure 3). Comparison between the DMVD and DMVD with PH groups showed that only the number of TPH-1- positive pneumocytes was higher in the DMVD with PH group (Figure 3). As for downstream signaling proteins, ERK1/2- and pERK1/2- positive pneumocytes were very low in the control group. Although ERK1/2- positive pneumocytes were significantly increased in the DMVD and DMVD with PH groups compared with the control group, pERK1/2- positive pneumocytes remained constant (Figure 3 and Figure 4). According to PAMs, only SERT- and 5-HTR2A-positive cells were increased in the DMVD and DMVD with PH groups compared with the control group. The numbers of PAMs expressing TPH-1, ERK1/2, and pERK1/2 proteins was stable in all groups (Figure 2, Figure 3 and Figure 4). Immunostaining of TPH-1, SERT, 5-HTR2A, and ERK1/2 was detected in bronchial epithelial cells of all dogs, with no significant difference among groups, whereas the expression of pERK1/2 in bronchial epithelial cells was absent in the control group, low in the DMVD group, and high in the DMVD with PH group (Figure 3).

### 3.4. The Correlation between the Relative Expression and Echocardiographic Parameters

In lung tissue, the relative expression of the *tph1* was positively correlated with LA/Ao (*r* = 0.648, *p* = 0.043). Moreover, the relative expression of pERK1/2 was positively correlated with LA/Ao (*r* = 0.758, *p* = 0.011) and LVIDsN (*r* = 0.721, *p* = 0.019). In PA tissue, the relative *slc6a4* expression was positively correlated with LVIDdN (*r* = 0.857, *p* = 0.014) and LVIDsN (*r* = 0.857, *p* = 0.014). However, the negative correlation between PAP and 5-HTR2A (*r* = −0.943, *p* = 0.005) were found. The relative 5-HTR2A expression was negatively associated with IVSdN (*r* = −0.693, *p* = 0.026) in pneumocytes, whereas the relative expression of pERK1/2 was positively correlated with LVPWsN (*r* = 0.638, *p* = 0.047), and TPH-1 was positively correlated with %FS (*r* = 0.705, *p* = 0.023) and PAP (*r* = 0.829, *p* = 0.042). In PAM, the relative pERK1/2 expression was positively correlated with LA (*r* = 0.721, *p* = 0.019) and LA/Ao (*r* = 0.685, *p* = 0.029). In addition, the relative expression of ERK1/2 and SERT was positively correlated with LVPWsN (*r* = 0.648, *p* = 0.043) and PAP (*r* = 0.829, *p* = 0.042), respectively. The significant correlation between all relative expression and the echocardiographic parameters is presented in Table 7.

## 4. Discussion

Upregulation of proteins involved in the 5-HT pathway has been demonstrated in human patients and animal models with PH [32,41,42,43], suggesting that the 5-HT pathway is involved in the pathogenesis of PH. The results showed that the relative expression of genes and proteins related to the 5-HT pathway was not different in PA and lung tissues from normal and DMVD dogs with and without PH. The patterns of protein expression were different between lung and PA tissues. In addition, immunohistochemical staining of lung tissue revealed the expression of proteins related to the 5-HT pathway in several cell types.

In previous studies in other animal species, variation in gene expression related to the 5-HT pathway has been noted in both lung and PA tissues. In rat models, *tph1* expression was found to be dependent on the different PH phenotypes, with *tph1* upregulated in idiopathic pulmonary arterial hypertension, while expression did not change in hypoxic PH rats compared to normal rats [44]. The different *slc6a4* were also revealed in human patients. Up-regulation of *slc6a4* was detected in primary PH and pulmonary veno-occlusive disease, while *slc6a4* expression did not change in secondary PH, caused by the various diseases, compared with control patients [45]. No difference in *htr2a* expression was observed in mouse and rat models with PH [46,47]. In the present study, the expression of *tph1*, *slc6a4*, and *htr2a* genes related to the 5-HT pathway was detected in lung and PA tissues in control dogs and DMVD dogs with and without PH. However, the expression pattern could not be defined. Because gene expression does not necessarily reflect protein expression, further investigation of protein expression by Western blot analysis and immunohistochemistry was performed.

Remodeling of the wall of PA including medial thickening has been noted in patients with PH [48]. The studies in laboratory animals have shown that 5-HT stimulated vasoconstriction and remodeling by causing contraction, proliferation and differentiation of PASMC [27,28,29]. The medial thickening associated with the expression of proteins related to the 5-HT signaling of PA was highlighted in a previous immunohistochemical study of PA in dogs with DMVD and dogs with PH due to DMVD [33]. Similarly, the present study found that the relative expression of some proteins, TPH-1 and 5-HTR2A, tended to be upregulated in the PA tissues of dogs in the DMVD and DMVD with PH groups, although this did not reach statistical significance. Therefore, 5-HT may play a crucial role in PA wall remodeling in dogs with DMVD.

In lung tissue from PH-induced animal models, upregulation of proteins associated with the 5-HT pathway has been detected [49]. However, to the authors’ knowledge, expression of these proteins in canine lung tissue has not been reported. In this study, Western blot analysis and immunohistochemistry were performed to determine protein expression and to identify the cell types expressing these proteins. Although the relative expression of proteins involved in the 5-HT signaling did not differ significantly among groups by Western blot analysis, immunohistochemistry revealed upregulation of these proteins in the DMVD and DMVD with PH groups compared with the control group.

Several publications have demonstrated that TPH-1 was associated with PH [44,49,50,51]. Although 5-HT is mainly synthesized in the gastrointestinal tract, local synthesis of 5-HT has been reported in the lung of mice [52]. The TPH-1 expression was upregulated in the lung tissue of PH-induced mice compared to controls [49,53]. A similar result was found in this study, which showed increased expression of TPH-1 in the lung tissue of dogs affected with DMVD with and without PH. The expression was highest in the DMVD with PH group. Overexpression of TPH-1 was found in pneumocytes along the alveolar wall, suggesting that pneumocytes may be one of the sources of 5-HT synthesis in the lung tissue of DMVD dogs with PH. However, neither the levels of 5-HT nor the expression of 5-HT in lung tissue were examined in this study, and further studies should be performed.

The 5-HT is transported via the PAECs and PASMCs through the mediation of SERT [18,54]. In a previous immunohistochemical study of dogs with PH due to DMVD, SERT was expressed mainly in PASMCs, and upregulation of SERT was detected in dogs affected with DMVD with and without PH compared with normal dogs [33]. In lung tissue, the immunohistochemical study showed that SERT was expressed by pneumocytes and PAMs, similar to an investigation in rat models [49,55,56]. The present study showed upregulation of SERT in dogs affected with DMVD with and without PH compared with normal dogs, but there was no significant difference between DMVD dogs with and without PH. It was hypothesized that the upregulation of SERT in pneumocytes may be responsible for the release of 5-HT in lung tissue.

The effects of 5-HT are mediated not only via SERT but also via 5-HT receptors. There are several types of serotonin receptors that related to PH including 5-HTR2A, 5-HT 5-HTR1B and 5-HTR2B [57]. Several studies have provided evidence that the 5-HTR2A plays an important role in pulmonary diseases including PH [58,59,60,61,62], and 5-HT2A has been reported to mediate pulmonary arterial remodeling in rats [58,59]. In dogs, 5-HT2A receptors were expressed in the coronary arteries and mediated the effects of 5-HT induced vasoconstriction [63]. Therefore, 5-HTR2A was selected for this study. Western blot analysis showed that the relative 5-HTR2A expression in lung and PA tissues did not differ among groups; similar results have been found in lung and PA tissues from humans, mice and rats [45,46,47]. Immunolocalization of 5-HTR2A in lung tissue was found in pneumocytes, PAMs and bronchial epithelial cells of all recruited dogs suggesting that 5-HTR2A may act on many cell types in the lung. Overexpression of 5-HTR2A was found in pneumocytes and PAMs from dogs affected with DMVD with and without PH compared with normal dogs. A similar picture was found in the studies of mice induced chronic obstructive pulmonary disease and in human patients with idiopathic pulmonary fibrosis (IPF) [60,61,62]. Previous studies have shown that not only does 5-HTR2A have an effect on PH, but that other 5-HT receptors also play a crucial role in PH. For example, the 5-HTR1B and the 5-HTR2B have been implicated in PH in piglets and mice [64,65,66], and 5-HTR2A, 5-HTR2B, 5-HT receptor 7, and SERT have been involved in pulmonary vascular remodeling in rats with PH [67,68,69]. Further studies to clarify the cooperative function between SERT and the 5-HT receptor in dogs with PH should be conducted.

Both ERK1/2 and pERK1/2 are downstream effectors that mediate proliferation and inhibit apoptosis of PASMCs [27,29,70]. Previous investigations have demonstrated that an upregulation in ERK1/2 and pERK1/2 protein expression reflects 5-HT that influence the enhanced proliferation and decreased apoptosis of PASMCs via SERT and 5-HTR2A [27,28,29,70]. A previous immunohistochemical study in canine PA showed increased ERK1/2 and pERK1/2 expression in DMVD dogs, but only pERK1/2 increased in dogs with PH due to DMVD, suggesting that the 5-HT-ERK1/2 signaling pathway appears to be stimulated in dogs affected with DMVD and PH and is related to PA remodeling [33]. In the present study, the expression of ERK1/2 and pERK1/2 in lung tissues, as determined by immunohistochemistry, was upregulated in the pneumocytes of DMVD dogs with and without PH compared with controls, but only ERK1/2 reached a significant level. Overexpression of ERK1/2 and pERK1/2 under hypoxic conditions has been reported in several cells [53,71,72,73]. It would be possible that increased ERK1/2 and pERK1/2 expression occurred in pneumocytes from dogs affected with DMVD with and without PH occurred secondary to hypoxia due to pulmonary edema. Surprisingly, ERK1/2 expression was significantly reduced in dogs with PH compared with those without PH. DMVD dogs with PH received more aggressive treatments with cardiovascular drugs including sildenafil. Investigation of the effect of sildenafil on PASMCs revealed that sildenafil can inhibit ERK1/2 expression and promote the degradation of pERK1/2 [74]. In addition, sildenafil can improve peak oxygen consumption and exercise capacity in PH patients secondary to heart failure [75,76,77,78,79,80]. In PH dog secondary to respiratory disease, sildenafil can improve respiratory rate and radiographic findings [81]. The reduction of ERK1/2 in the DMVD with PH group may have been caused by the use of sildenafil. However, because only one dog in this group was prescribed sildenafil, the other causes that might affect the reduction of ERK1/2 expression in the DMVD with PH group should be further investigated. In contrast to the PASMCs, pERK1/2 tended to be increase in dogs affected with DMVD with and without PH but did not reach a significant level. This may be due to the following: (1) the sample size in this study was smaller than in previous studies, (2) pneumocytes may be a source not an effector cell of 5-HT in lung tissue, and (3) ERK1/2 may not be the major effector proteins of the 5-HT-signaling pathway in the lungs of dogs affected with DMVD with and without PH. Other effector proteins, such as ROCK and MAPK, may be involved in the 5-HT signaling in the lungs of DMVD dogs, requiring further investigation for clarification.

The present study has shown that the relative expression of some targeted genes and proteins in lung, PA, pneumocytes, and PAM was strongly correlated with some echocardiographic parameters. The positive correlation between the 5-HT-effector expression in all targeted tissues and cells and LA/Ao, LVIDdN, and PAP may support that an increase in the activation of the 5-HT signaling pathway can induce more echocardiographic changes in DMVD dogs with and without PH. The studies in animal models with induced PH mentioned that blockage of SERT and 5-HT receptors can decrease the cardiac structural changes of the and PAP [68,69,82]. A positive correlation between TPH-1 and %FS, pERK1/2 and LVPWsN was found in pneumocytes and between ERK1/2 and in PAM. These findings reflect the increase in %FS and LV wall thickening during disease progression. Progressive mitral regurgitation in DMVD increases cardiac workload, leading to LV remodeling [34]. In addition, all dogs in the DMVD and DMVD with PH groups were treated with pimobendane, which has been reported to induce ventricular hypercontractility leading to a higher %FS [83]. Being negatively correlated with PAP of PA tissue and IVSdN of pneumocytes, the relative 5-HTR2A expression was contrasted with the results. Working with the larger sample size may clarify this distinction and prove that correlation between echocardiographic parameters and relative expression of target genes and proteins may lead to the prediction of the changes in gene and protein expression.

This study has limitations, including the small sample size that may have affected the absence of a significant difference of the protein and gene expressions. A further study with a larger number of samples should be performed to clarify the significant changes in gene and protein expression of 5-HT mediators.

## 5. Conclusions

In summary, the present study shows that the expression of genes and proteins of the 5-HT-signaling was found in canine lung and PA tissues. The pattern of expression was different between these two locations. Therefore, the 5-HT pathway in lung and PA tissues might play different roles in DMVD with and without PH. The paracrine and autocrine effects of this pathway in lung and PA tissues should be further investigated to better understand the importance of the 5-HT pathway in the development of PH due to DMVD in dogs. The information obtained from this study could provide insightful information regarding the pathogenesis of PH secondary to DMVD in dogs and may inspire researchers to develop a further experimental study to treat or slow the progression of PH using selective inhibitors or drugs that can block the 5-HT pathway in the future.

## Figures and Tables

**Figure 1 vetsci-09-00530-f001:**
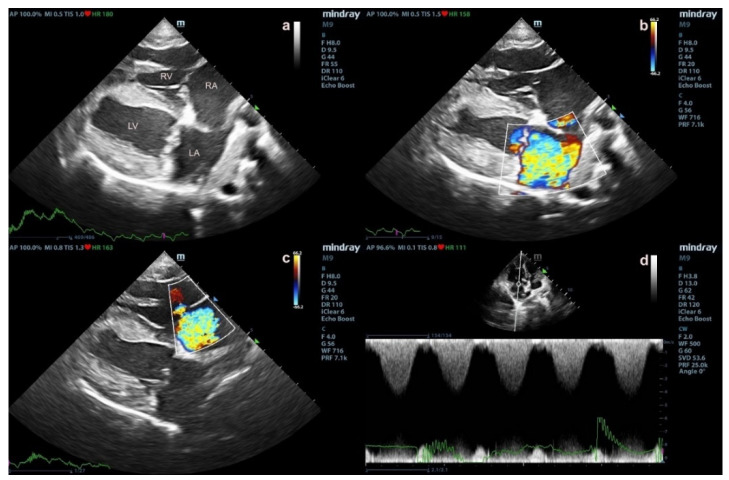
The echocardiographic findings of degenerative mitral valve disease (DMVD) dogs with pulmonary hypertension (PH) (**a**–**d**). Echocardiography from two-dimensional right parasternal four chamber view shows mitral valve thickening and left atrial (LA) enlargement (**a**). Color-flow Doppler echocardiography shows mitral valve (**b**) and tricuspid valve regurgitation jets (**c**). Spectral Doppler echocardiography shows peak systolic tricuspid regurgitation velocity (**d**).

**Figure 2 vetsci-09-00530-f002:**
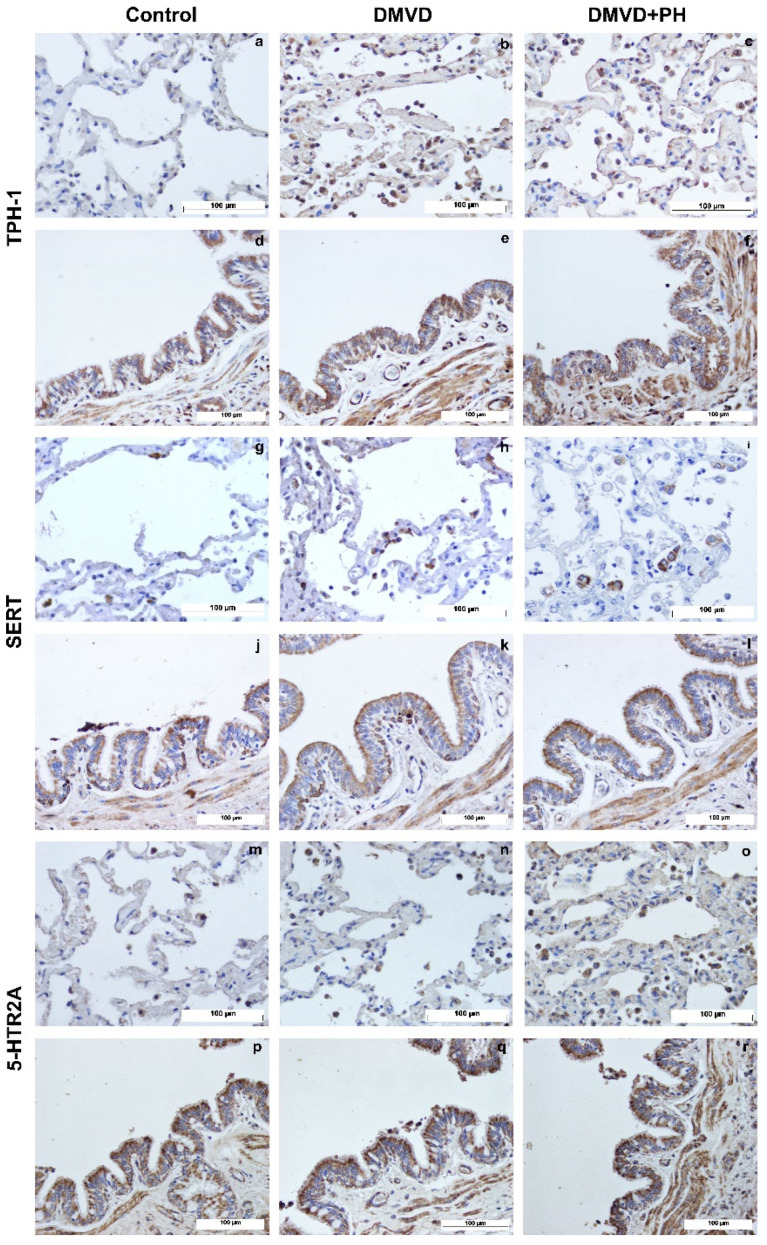
The cytoplasmic expression of TPH-1, SERT and 5-HTR2A in pneumocytes of alveoli PAMs (top row) and bronchial epithelial cells (bottom row) of dogs in the control (**a**,**d**,**g**,**j**,**m**,**p**), degenerative mitral valve disease (DMVD) (**b**,**e**,**h**,**k**,**n**,**q**) and degenerative mitral valve disease with pulmonary hypertension (DMVD with PH) groups (**c**,**f**,**i**,**l**,**o**,**r**) presented as brown color (Labeled streptavidin-biotin, Immunohistochemistry, Mayer’s Hematoxylin counterstained, 400× magnification).

**Figure 3 vetsci-09-00530-f003:**
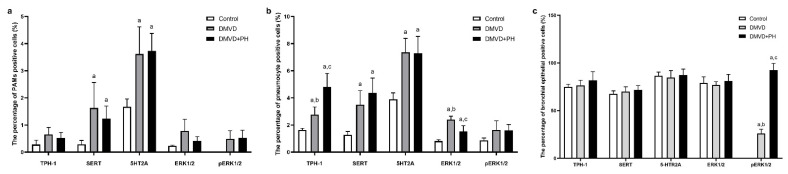
The average percentage of TPH-1, SERT, 5-HTR2A, ERK1/2 and pERK1/2 positive pneumocytes (**a**), PAMs (**b**) and bronchial epithelial cells (**c**) in the control, degenerative mitral valve disease (DMVD) and degenerative mitral valve disease with pulmonary hypertension (DMVD with PH) groups. Data are presented as mean and standard deviation (bars). ^a^ Indicate significant difference at *p* < 0.05 compared to the control group. ^b,c^ Indicate significant difference at *p* < 0.05 between the DMVD and DMVD with PH groups.

**Figure 4 vetsci-09-00530-f004:**
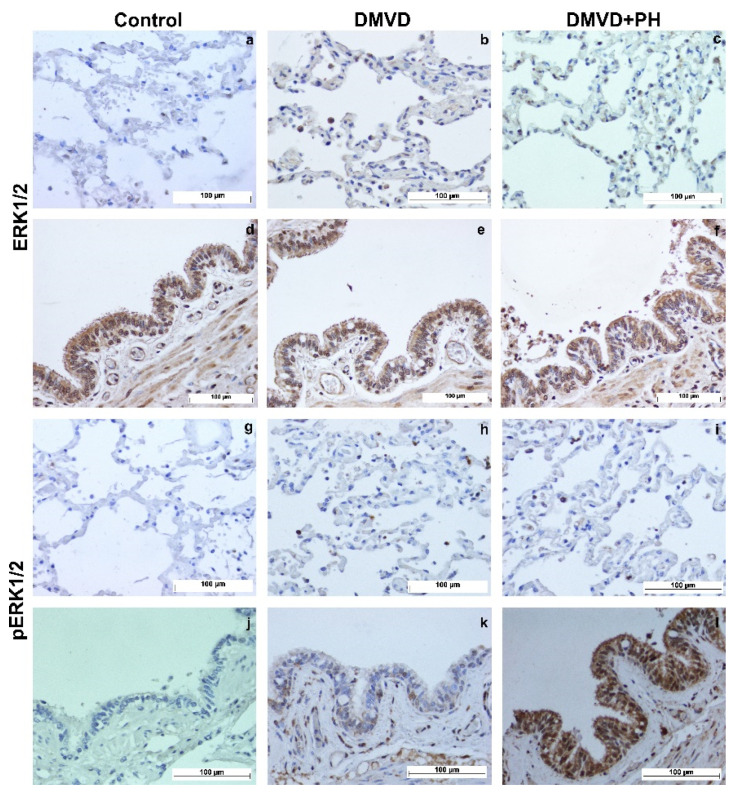
The expression of ERK1/2 and pERK1/2 in pneumocytes, PAMs (top row) and bronchial epithelial cells (bottom row) of dogs in the control (**a**,**d**,**g**,**j**), degenerative mitral valve disease (DMVD) (**b**,**e**,**h**,**k**) and degenerative mitral valve disease with pulmonary hypertension (DMVD with PH) groups (**c**,**f**,**i**,**l**) presented as brown color (Labeled streptavidin-biotin, Immunohistochemistry, Mayer’s Hematoxylin counterstained, 400× magnification).

**Table 1 vetsci-09-00530-t001:** Primers for qRT-PCR associating serotonin signaling pathway.

Genes	Accession No.	Forward Primer (5′ to 3′)	Reverse Primer (5′ to 3′)	Product Size (bp)	Tm (°C)	Reference of Primers
*tph1*	NM_001197191.1	CTGTGGAGTTTGGTCTCTGTAAG	TGTGATGAGACACTCCTGTTTG	158	82.77	This study
*slc6a4*	NM_001110771.1	GGCTGAGATGAGGAACGAAG	TTGGACCAGATGTGTGGAAA	222	84.43	This study
*htr2a*	NM_001005869.1	TCTTTCAGCTTCCTCCCTCA	TCCTCGTTGCAGGACTCTTT	227	84.70	This study
*RPS19*	XM_005616513.3	CCTTCCTCAAAAAGTCTGGG	GTTCTCATCGTAGGGAGCAAG	95	80.69	Brinkhof et al. [39]
*RPL32*	XM_022406256.1	TGGTTACAGGAGCAACAAGAAA	GCACATCAGCAGCACTTCA	100	81.54	Peters, Peeters, Helps and Day [37]

**Table 2 vetsci-09-00530-t002:** Signalment of dogs in the normal, degenerative mitral valve disease (DMVD) and degenerative mitral valve disease with pulmonary hypertension (DMVD with PH) groups.

	Normal (*n* = 4)	DMVD (*n* = 5)	DMVD with PH (*n* = 5)	*p*-Value
Sex (male/female)	3/1	1/4	3/2	-
Age (years)	8 (7.8–10.0)	15 (14–15)	14 (13–15)	
Weight (kg)	5.6 (5.28–6.93)	3.92 (3.7–4.14)	4.3 (2.8–6.9)	
Breed	Shih-Tzu (*n* = 2) Cocker Spaniel (*n* = 1) Mixed breed (*n* = 1)	Poodle (*n* = 3) Shih-Tzu (*n* = 1) Pomeranian (*n* = 1)	Chihuahua (*n* = 2) Poodle (*n* = 2) Pomeranian (*n* = 1)	-
Stage (C/D)	-	2/3	1/4	-
Medication	-	ACEIs (*n* = 5) Furosemide (*n* = 5) Pimobendane (*n* = 5) Spironolactone (*n* = 2) Moduretic (*n* = 1)	ACEIs (*n* = 5) Furosemide (*n* = 5) Pimobendane (*n* = 5) Spironolactone (*n* = 2) Moduretic (*n* = 2) Sildenafil (*n* = 1)	-

The data are expressed as median (interquartile range). The *p* value was analyzed by Kruskal-Wallis test.

**Table 3 vetsci-09-00530-t003:** Echocardiographic data of dogs in degenerative mitral valve disease (DMVD) and degenerative mitral valve disease with pulmonary hypertension (DMVD with PH) groups.

	DMVD (*n* = 5)	DMVD with PH (*n* = 5)	*p*-Value
LA index (cm/kg)	1.67 ± 0.28	1.83 ± 0.72	0.642
Ao index (cm/kg)	1.01 ± 0.42	1.02 ± 0.28	0.952
LA/Ao	1.80 ± 0.56	1.75 ± 0.38	0.862
IVSdN index (cm/kg)	0.59 ± 0.13	0.56 ± 0.14	0.663
LVIDdN index (cm/kg)	2.84 ± 0.87	1.66 ± 0.62	0.038
LVPWdN index (cm/kg)	0.55 ± 0.24	0.48 ± 0.17	0.621
IVSsN index (cm/kg)	0.92 ± 0.30	0.84 ± 0.29	0.707
LVIDsN index (cm/kg)	1.59 ± 0.50	0.72 ± 0.30	0.010
LVPWsN index (cm/kg)	0.85 ± 0.33	0.71 ± 0.10	0.407
%FS	42.14 ± 5.82	55.16 ± 7.91	0.018
PAP (mmHg)	-	72.03 ± 25.96	-

Abbreviation: Ao = aorta; FS = fractional shortening; IVSdN = normalised interventricular septal thickness at end diastole; IVSsN = normalised interventricular septal thickness at end systole; LA = left atrium; LA/Ao = left atrium to aorta ratio; LVIDdN = normalised left ventricular internal diameter at end diastole; LVIDsN = normalised left ventricular internal diameter at end systole; LVPWdN = normalised left ventricular posterior wall thickness at end diastole; LVPWsN = normalised left ventricular posterior wall thickness at end systole; PAP = pulmonary arterial pressure. Data are expressed as mean ± standard deviation. *p*-values indicate the significant difference between two groups by the pindependent *t*-test.

**Table 4 vetsci-09-00530-t004:** The relative expression of the targeted genes.

Gene	Normal (*n* = 4)	DMVD (*n* = 5)	DMVD with PH (*n* = 5)	*p*-Value
Lung tissues (×10^−3^)				
*tph1*	0.61 (0.04–3.29)	0.24 (0.06–4.21)	0.13 (0.07–0.53)	0.680
*slc6a4*	3.32 (0.19–6.23)	1.11 (0.58–3.03)	0.27 (0.13–1.03)	0.081
*htr2a*	0.33 (0.23–0.68)	0.10 (0.09–0.55)	0.33 (0.29–0.34)	0.154
PA tissues (×10^−3^)				
*tph1*	0.20 (0.04–3.89)	1.89 (0.58–6.58)	1.01 (0.45–1.71)	0.117
*slc6a4*	1.60 (0.08–5.89)	11.03 (10.66–11.40)	0.74 (0.12–14.94)	0.252
*htr2a*	14.46 (0.85–20.05)	2.00 (0.43–3.22)	5.79 (0.41–19.19)	0.270

Abbreviation: *tph1* = tryptophan hydroxylase 1 gene; *slc6a4* = solute carrier family 6 member 4 (serotonin transporter gene); *htr2a* = 5-hydroxytryptamine receptor 2A gene. Data are expressed as median (minimum-maximum range). *p*-values indicate the significant difference among three groups by the Kruskal-Wallis test.

**Table 5 vetsci-09-00530-t005:** The relative protein expression.

Proteins	Normal (*n* = 4)	DMVD (*n* = 5)	DMVD with PH (*n* = 5)	*p*-Value
Lung tissues				
TPH-1	2.33 (0.58–4.09)	1.59 (0.52–5.43)	1.56 (0.68–5.08)	0.362
SERT	1.14 (0.09–3.10)	0.46 (0.04–1.00)	1.49 (0.24–1.69)	0.373
5-HTR2A	4.40 (0.97–10.75)	2.59 (1.12–9.46)	3.15 (2.36–8.88)	0.656
ERK1/2	1.74 (0.68–6.37)	1.41 (0.72–5.38)	2.16 (1.04–4.29)	0.680
pERK1/2	0.72 (0.13–2.01)	1.02 (0.35–4.09)	0.21 (0.07–2.59)	0.284
PA tissues				
TPH-1	0.65 (0.49–1.12)	0.81 (0.29–4.09)	1.85 (0.51–2.95)	0.401
SERT	0.23 (0.14–0.39)	0.14 (0.04–0.63)	0.40 (0.04–0.74)	0.761
5-HTR2A	0.68 (0.44–1.00)	0.99 (0.37–4.67)	1.58 (0.50–2.57)	0.350
ERK1/2	0.86 (0.73–1.04)	0.81 (0.54–4.23)	1.04 (0.48–4.15)	0.831
pERK1/2	0.61 (0.35–0.89)	0.45 (0.11–2.26)	0.56 (0.10–2.87)	0.990

Abbreviation: TPH-1 = tryptophan hydroxylase-1; SERT = serotonin transporter; 5-HTR2A = serotonin receptor 2A; ERK1/2 = extracellular signal-regulated kinase ½; pERK1/2 = phosphorylated extracellular signal-regulated kinase ½. Data are expressed as median (minimum-maximum range). *p*-values indicate the significant different among three groups analyzed by the Kruskal-Wallis test.

**Table 6 vetsci-09-00530-t006:** The average percentage of TPH-1, SERT, 5-HTR2A, ERK1/2 and pERK1/2 positive pneumocytes, PAMs and bronchial epithelial cells in the control, degenerative mitral valve disease (DMVD) and degenerative mitral valve disease with pulmonary hypertension (DMVD with PH) groups.

Proteins	Cells	Normal (*n* = 4)	DMVD (*n* = 5)	DMVD with PH (*n* = 5)	*p*-Value
TPH-1	Pneumocytes	1.63 ± 0.13	2.76 ± 0.57 ^a,b^	4.90 ± 0.99 ^a,c^	<0.001
	PAMs Bronchial epithelial cells	0.28 ± 0.16 74.84 ± 2.82	0.65 ± 0.26 76.62 ± 5.33	0.52 ± 0.21 81.83 ± 8.98	0.074 0.273
SERT	Pneumocytes	1.28 ± 0.24	3.50 ± 1.03 ^a^	4.35 ± 1.11 ^a^	<0.001
	PAMs Bronchial epithelial cells	0.29 ± 0.14 67.59 ± 3.12	1.63 ± 0.94 ^a^ 69.87 ± 5.09	1.23 ± 0.47 ^a^ 71.90 ± 4.37	0.028 0.371
5-HTR2A	Pneumocytes	3.89 ± 0.48	7.36 ± 1.03 ^a^	7.29 ± 1.24 ^a^	<0.001
	PAMs Bronchial epithelial cells	1.67 ± 0.29 86.52 ± 3.90	3.62 ± 1.00 ^a^ 84.79 ± 7.24	3.73 ± 0.65 ^a^ 87.32 ± 6.24	0.005 0.804
ERK1/2	Pneumocytes	0.81 ± 0.10	2.40 ± 0.25 ^a,b^	1.54 ± 0.41 ^a,c^	<0.001
	PAMs Bronchial epithelial cells	0.24 ± 0.02 78.84 ± 6.55	0.78 ± 0.43 76.82 ± 3.51	0.41 ± 0.16 81.01 ± 6.95	0.059 0.540
pERK1/2	Pneumocytes	0.86 ± 0.18	1.63 ± 0.68	1.61 ± 0.43	0.243
	PAMs Bronchial epithelial cells	0.00 ± 0.00 0.00 ± 0.00	0.49 ± 0.30 26.16 ± 4.39 ^a,b^	0.53 ± 0.28 92.39 ± 7.19 ^a,c^	0.108 < 0.001

Abbreviation: TPH-1 = tryptophan hydroxylase-1; SERT = serotonin transporter; 5-HTR2A = serotonin receptor 2A; ERK1/2 = extracellular signal-regulated kinase 1/2; pERK1/2 = phosphorylated extracellular signal-regulated kinase 1/2. Data are expressed as mean and standard deviation. *p*-values <0.5 indicate different among groups was analyzed by one way ANOVA test. ^a^ Indicate statistically significant difference at *p* < 0.05 compared to the control group. ^b,c^ Indicate statistically significant difference at *p* < 0.05 between the DMVD and DMVD with PH groups.

**Table 7 vetsci-09-00530-t007:** The significant Spearman correlation between the relative expression of the targeted genes and proteins and the echocardiographic parameters.

Organ or Cell	Genes or Proteins	Echo Parameters	*r*	*p*-Value
Lung	*tph1*	LA/Ao	0.648	0.043
	pERK1/2	LA/Ao	0.758	0.011
	pERK1/2	LVIDsN	0.721	0.019
PA	*slc6a4*	LVIDdN	0.857	0.014
	*slc6a4*	LVIDsN	0.857	0.014
	5-HTR2A	PAP	−0.943	0.005
Pneumocyte	TPH-1	%FS	0.705	0.023
	TPH-1	PAP	0.829	0.042
	5-HTR2A	IVSdN	−0.693	0.026
	pERK1/2	LVPWsN	0.638	0.047
PAM	SERT	PAP	0.829	0.042
	ERK1/2	LVPWsN	0.648	0.043
	pERK1/2	LA	0.721	0.019
	pERK1/2	LA/Ao	0.685	0.029

Abbreviation: *tph1* = tryptophan hydroxylase 1 gene; TPH-1 = tryptophan hydroxylase-1; *slc6a4* = solute carrier family 6 member 4 (serotonin transporter gene); SERT = serotonin transporter; 5-HTR2A = serotonin receptor 2A; ERK1/2 = extracellular signal-regulated kinase 1/2; pERK1/2 = phosphorylated extracellular signal-regulated kinase 1/2; FS = fractional shortening; IVSdN = normalised interventricular septal thickness at end diastole; LA/Ao = left atrium to aorta ratio; LVIDdN = normalised left ventricular internal diameter at end diastole; LVIDsN = normalised left ventricular internal diameter at end systole; LVPWsN = normalised left ventricular posterior wall thickness at end systole; PAP = pulmonary arterial pressure.

## Data Availability

Not applicable.

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
