# Peer review of "Investigation of Genes and Proteins Expression Associating Serotonin Signaling Pathway in Lung and Pulmonary Artery Tissues of Dogs with Pulmonary Hypertension Secondary to Degenerative Mitral Valve Disease: The Preliminary Study"

_vetsci, 2022, doi:10.3390/vetsci9100530_

Round 1
Author Response
Please reduce the Similarity rate of the manuscript
(please reduce it to under 30%);
Please see the attached file of turn it in. Almost all of the similarity is terminology, name of products and companies, breeds of dogs, names of signaling proteins and statistical methods. It’s quite difficult to decrease the % of similarity to under 30%.
Reviewer 1
Comments and Suggestions for Authors
Type of manuscript: Article
Title: Investigation of genes and proteins expression associating serotonin signaling pathway in lung and pulmonary artery tissues of dogs with pulmonary hypertension secondary to degenerative mitral valve disease
Journal: Veterinary Sciences
General Comments:
 This article is the investigation to expression of 5-HT related protein and genes from samples of clinical DVMD dogs with/without PH. The aim of this study is very interesting to veterinarian and cardiologist. Although the research is very informative and useful, I feel that the author’s opinions are not reflected in the discussion section. Some revisions are expected to be more excellent articles.
Major Comments:
- Line 255-262 and Table 5. In western blot analysis, authors explain the trend of the protein expression. However, SERT, ERK 1/2 and pERK1/2 were not upregulated in the DMVD. These proteins had spread range of values, in graphic, these proteins may be showed the trend. In table 5, these values were showed as median. I could not understand “the trend of dogs in the DMVD and DMVD+PH groups”. Please revise the sentences or table values in visually.
- The mentioned context, “but these proteins tended to be upregulated in the PA tissues of dogs in the DMVD and DMVD+PH groups”. Was corrected as follows.
“In lung tissue, the expression patterns for TPH-1, SERT, 5-HTR2A, ERK1/2 and pERK1/2 proteins could not be determined, but TPH-1 and 5-HTR2A tended to be upregulated in the PA tissues of dogs in the DMVD and DMVD+PH groups. In addition, the expression of SERT and ERK1/2 tended to be upregulated in the DMVD+PH groups.”
- Line 351, please include the aims of this study in discussion. It was difficult to understand without explanation of the purpose between background and results.
- The objective of the study was included as follows.
“Consequently, the present study aimed to investigate the gene and protein expressions reflecting the local signalling of the 5-HT pathway in lung and PA tissues of DMVD dogs with PH compared with normal dogs and DMVD dogs without PH. The results of this study showed that the relative expression of genes and proteins related to the 5-HT pathway was not different in PA and lung tissues from normal and DMVD dogs with and without PH.”
- Line 357-, Is this all there is to the discussion on gene expression?
- More discussion was added before the mentioned context to shoe the variation of gene expression in line 351-364 as follows.
“The previous studies in other species found the variation in gene expression associating 5-HT pathway in both lung and PA tissues. In rat models, the tph1 expression was found depending on the different PH phenotypes which the tph1 was upregulated in idiopathic pulmonary arterial hypertension while the expression in hypoxic PH rats was not change compared with the normal rats [45]. The various slc6a4 was also revealed in human patients. The slc6a4upregulation was found in primary PH and pulmonary veno-occlusive disease while the slc6a4 expression in secondary PH caused by the various diseases was not change compared with the control patients [46]. No difference in htr2a expression was observed in mouse and rat models with PH [47,48]. In the present study, the expression of tph1, slc6a4 and htr2a genes related to the 5-HT pathway was detected in lung and PA tissues in control dogs and DMVD dogs with and without PH. However, the expression pattern could not be defined. Because gene expression does not necessarily reflect protein expression, further investigation of protein expression by Western blot analysis and immunohistochemistry was performed.”
- Line 362-367, from the results of this study and previous study, how did you consider this? This paragraph was only explanation.
- We had discussed more details to describe the association of 5-HT and PA remodelling in line 365-374 as follows.
“The remodelling of PA wall including medial thickening was found in patient with PH [49]. The studies in laboratory animals indicated that 5-HT stimulated vasoconstriction and remodelling by causing PASMC contraction, proliferation and differentiation [27-29]. The medial thickening associated with the expression of proteins related to the 5-HT pathway of canine PA was highlighted in a previous immunohistochemical study of PA in dogs with DMVD and dogs affected by PH secondary to DMVD [33]. Similarly, the present study found that the relative expression of some proteins, TPH-1 and 5-HTR2A, tended to be upregulated in the PA tissues of dogs in the DMVD and DMVD+PH groups, although it did not reach statistical significance. Therefore, 5-HT may have a crucial role in PA wall remodelling of dogs with DMVD.”
- Line 384-385 and Line 395-396, these sentences mean the same limitation of this study.
- The mentioned context in line 395-396 was removed.
- Line 413-416, What are the implications of these references for this study? In my opinion, as 5-HTR2A is associated with different types of PH, it would be interesting if different receptors were expressed in different PH categories.
- We would like to mention that other 5-HT receptors also affect PH, so we insert the previous studies in various 5-HT receptors of other species to emphasize this point. For prevention of the misunderstanding of these contexts, the mentioned details were rearranged as follows from “Previous studies have shown that not only 5-HTR2A has an effect on PH, but also other 5-HT receptors play a crucial role in PH. The 5-HTR1B and the 5-HTR2B have been implicated in PH in piglets and mice [64-66]. Inhibition of 5-HTR2A, 5-HTR2B, 5-HT receptor 7 and SERT in rats with PH attenuated pulmonary vascular remodelling [67-69].” to “Previous studies have shown that not only 5-HTR2A has an effect on PH, but also other 5-HT receptors play a crucial role in PH. For example, the 5-HTR1B and the 5-HTR2B have been implicated in PH in piglets and mice [64-66], and 5-HTR2A, 5-HTR2B, 5-HT receptor 7 and SERT have been involved in pulmonary vascular remodelling in rats with PH [67-69].”
- Line 422-423, Please include more information. ERK1/2 was significantly increased in DMVD dogs. Although pERK1/2 was increased in PH dogs in previous study, pERK1/2 was increased in both DMVD groups in this study. However, this fact was not discussed, and the expression of ERK1/2 was perceived as unimportant. In my opinion, is there any difference from previous studies in the severity of PH or in the deterioration of specimens over time?
- Previous study performed immunohistochemical expression of ERK1/2 and pERK1/2 in the PA of dogs in the DMVD and DMVD+PH groups compared to the healthy control group and found that ERK1/2 was significantly increased in the PASMC of dogs in the DMVD and DMVD+PH groups while pERK1/2 was significantly increased only in the DMVD+PH group compared to the control group. This present study also used the same samples from previous study to evaluate the expression of ERK1/2 and pERK1/2 in lung tissues. The results showed that the expression pattern of downstream effectors, ERK1/2 and pERK1/2 were different from previous study. ERK1/2 was significantly increased in the pneumocytes of dogs in the DMVD and DMVD+PH groups compared to the control group and it was highest in the DMVD group and significantly decreased in the DMVD+PH group. Whereas pERK1/2 was trend to increase in the DMVD and DMVD+PH groups compared to the control group but not reached a significant level.
- The more discussion was added in line 445-466 as follows:
The overexpression of ERK1/2 and pERK1/2 in hypoxic condition has been reported in several cells [54,72-74]. It could have been possible that an increased expression of ERK1/2 and pERK1/2 in pneumocytes of DMVD dogs with and without PH occurred secondary to hypoxia from pulmonary edema. Surprisingly, ERK1/2 expression was significantly reduced in the DMVD+PH group compared to the DMVD group. Dogs in the DMVD+PH group were received more aggressive cardiovascular drugs treatment including sildenafil. The study effect of sildenafil on PASMCs revealed that sildenafil can inhibit ERK1/2 expression and promote degradation of pERK1/2 [75]. Moreover, sildenafil can improve peak oxygen consumption and exercise capacity in PH secondary to heart failure patients [76-81]. In PH dog secondary to respiratory diseases, sildenafil can improve respiratory rate and radiographic findings [82]. Reduction of ERK1/2 in the DMVD+PH group might have been caused by the used of sildenafil. However, only one dogs in this group were prescribed with sildenafil, the other causes that may affected the reduction of ERK1/2 expression in the DMVD+PH group should be further identified. Unlike in PASMCs, pERK1/2 was trend to increase in the DMVD and DMVD+PH groups but not reached a significant level. The possible suggestion are as follows: 1) the sample size in this present study was smaller than previous study, 2) the pneumocytes may be a source not an effector cell of 5-HT in lungs tissues, 3) ERK1/2 may not be the major effector proteins of the 5-HT-signaling pathway in the lungs of DMVD dogs with and without PH. Other effector proteins, such as ROCK and MAPK, may play a role in the 5-HT pathway in the lungs of DMVD dogs, which requires further investigation for clarification.
Minor Comments:
- Line 49, “PASMCs” have been already indicated.
- The abbreviated word, PASMCs, was replaced in the sentence.
- Line 50-51, mitogen-activated protein kinase/extracellular signal-related kinase kinase (MEK/ERK)→mitogen-activated protein kinase kinase/extracellular signal-related kinase?
- mitogen-activated protein kinase/extracellular signal-related kinase kinase (MEK/ERK) was changed to mitogen-activated protein kinase kinase/extracellular signal-related kinase (MEK/ERK).
- Line 45-47, Is this sentence in the right position? It is a little confusing.
- The mentioned sentence is in the corrected position, but it is too short to describe the understanding. Therefore, we added some words to complete the sentence as follows. “5-HT was hypothesized that it can cause PH since 5-HT levels have found increase in women treated with SERT substrate anorectic drugs, resulting in an increased risk of PH [24,25].”
- Line 227, “and” is better to not italic.
- The italic word was canceled.
- Line 232-233, “DMVD dogs with and without PH” is better to change “DMVD and DMVD+PH groups”.
- “DMVD dogs with and without PH” was changed to “DMVD and DMVD+PH groups”.
- Table 4, what groups between were the p values indicated?
- The p value indicated the difference between all three groups. Therefore, we added more explanations in table 4 and 5 as follows “The p value indicating among three groups was analyzed by Kruskal-Wallis test.”
- Line 327, PAP “and” 5-HTR2A. Please add.
- “and” was added to the sentence, “However, the negative correlation between PAP and 5-HTR2A (r=-0.943, p=0.005) were found.”.
- Line 403, “5-HT2A” is “5-HTR2A”?
- 5-HT2A was replaced as 5-HTR2A
That is all of my comments.

Reviewer 2 Report
Dear Authors,
I read your manuscript with great interest. In my opinion, the topic you have raised has a potential of importance in veterinary cardiology, although due to small number of studied animals the conclusions should be drawn with great cautiousness.
In my opinion some corrections should be made in your manuscript before considering publication.
- as you mention, the study group is small, therefore the conclusions are difficult to be drawn; is there a possibility to enlarge the study group (especially DMVD and DMVD+PH groups)? If not, the title should include information that the study is preliminary
- the information presented in table 2 are also described in detail in the text (lines 203-213) leading to unnecessary repetition; the table alone with a short summary in the text would be sufficient
- line 260: the mentioned pattern is incorrect: only TPH-1 and 5-HTR2A proteins show a growing trend in DMVD and DMVD+PH groups while in other proteins the expression stays on similar level or even decrease
- lines 280-281: the ERK1/2 is significantly increased in DMVD and DMVD-PH group as compared to control group; nonetheless, you also show a significant decrease in ERK1/2 expression in DMVD-PH group as compared to DMVD group. Can you provide explanation for that phenomenon?
- lines 285-289: you state that the expression of TPH-1, SERT, 5-HTR2A, ERK1/2 was observed in bronchial cells in IHC examination; in my opinion, it should be presented in figure 2 and 3 together with the expression in alveoli and PAMs as for pERK1/2 even if no difference between groups was observed.
- Figure 4 and Table 6 should also contain the results of immunohistochemical examination of the proteins in bronchial cells
- line 327: correct the sentence - it should probably be "between PAP and 5-HTR2A"
- line 434: although you have shown the correlation with some echocardiographic parameters, no differences in the gene expression was shown between the groups (especially between DMVD and DMVD-PH groups) therefore the conclusion seems precipitative
- lines 432-444: the results of correlation between echocardiographic parameters and gene and protein expression are inconsistent and therefore the conclusions need more discussion to be drawn; tph1 and TPH-1 showed positive correlation with LA/Ao, %FS and PAP - while increase in LA/Ao and PAP is combined with worsening of the cardiac function, the increase in %FS is not unambiguous - the fractional shortening first increases to decrease in end-stage disease; similarly the pERK1/2 and ERK1/2 expression showed positive correlation with LVPWsN, LA and LA/Ao and as LA and LA/Ao increase is correlated with disease progression, the thickening of LV wall is not explicit.
Author Response
Reviewer 2
Dear Authors,
I read your manuscript with great interest. In my opinion, the topic you have raised has a potential of importance in veterinary cardiology, although due to small number of studied animals the conclusions should be drawn with great cautiousness.
In my opinion some corrections should be made in your manuscript before considering publication.
- as you mention, the study group is small, therefore the conclusions are difficult to be drawn; is there a possibility to enlarge the study group (especially DMVD and DMVD+PH groups)? If not, the title should include information that the study is preliminary
- The title was changed from “Investigation of genes and proteins expression associating serotonin signaling pathway in lung and pulmonary artery tissues of dogs with pulmonary hypertension secondary to degenerative mitral valve disease” to “Investigation of genes and proteins expression associating serotonin signaling pathway in lung and pulmonary artery tissues of dogs with pulmonary hypertension secondary to degenerative mitral valve disease: the preliminary study”
- the information presented in table 2 are also described in detail in the text (lines 203-213) leading to unnecessary repetition; the table alone with a short summary in the text would be sufficient
- The repetitive detail in the text was removed as follows.
“Fourteen canine cadavers were included in this study and divided into normal (n = 4), DMVD (n = 5) and DMVD+PH (n = 5) groups. The signalment including gender, age, body weight and breed of the enrolled dogs together with the stage of DMVD and the medication were shown in Table 2.”
- line 260: the mentioned pattern is incorrect: only TPH-1 and 5-HTR2A proteins show a growing trend in DMVD and DMVD+PH groups while in other proteins the expression stays on similar level or even decrease
- The mentioned context, “but these proteins tended to be upregulated in the PA tissues of dogs in the DMVD and DMVD+PH groups”. was corrected as follows.
“In lung tissue, the expression patterns for TPH-1, SERT, 5-HTR2A, ERK1/2 and pERK1/2 proteins could not be determined, but TPH-1 and 5-HTR2A tended to be upregulated in the PA tissues of dogs in the DMVD and DMVD+PH groups. In addition, the expression of SERT and ERK1/2 tended to be upregulated in the DMVD+PH groups.”
- lines 280-281: the ERK1/2 is significantly increased in DMVD and DMVD-PH group as compared to control group; nonetheless, you also show a significant decrease in ERK1/2 expression in DMVD-PH group as compared to DMVD group. Can you provide explanation for that phenomenon?
- The more discussion was added in line 449-459 as follows:
Surprisingly, ERK1/2 expression was significantly reduced in the DMVD+PH group compared to the DMVD group. Dogs in the DMVD+PH group were received more ag-gressive cardiovascular drugs treatment including sildenafil. The study effect of sildenafil on PASMCs revealed that sildenafil can inhibit ERK1/2 expression and pro-mote degradation of pERK1/2 [75]. Moreover, sildenafil can improve peak oxygen consumption and exercise capacity in PH secondary to heart failure patients [76-81]. In PH dog secondary to respiratory diseases, sildenafil can improve respiratory rate and radiographic findings [82]. Reduction of ERK1/2 in the DMVD+PH group might have been caused by the used of sildenafil. However, only one dogs in this group were prescribed with sildenafil, the other causes that may affected the reduction of ERK1/2 expression in the DMVD+PH group should be further identified.
- lines 285-289: you state that the expression of TPH-1, SERT, 5-HTR2A, ERK1/2 was observed in bronchial cells in IHC examination; in my opinion, it should be presented in figure 2 and 3 together with the expression in alveoli and PAMs as for pERK1/2 even if no difference between groups was observed.
- The picture of TPH-1, SERT, 5-HTR2A and ERK1/2 expression in bronchial epithelial cells were added in the figure 2 and 3.
- Figure 4 and Table 6 should also contain the results of immunohistochemical examination of the proteins in bronchial cells
- The results of immunohistochemical examination of TPH-1, SERT, 5-HTR2A, ERK1/2 and pERK1/2 were added in the Figure 4 and Table 6.
- line 327: correct the sentence - it should probably be "between PAP and 5-HTR2A"
- “and” was added to the sentence, “However, the negative correlation between PAP and 5-HTR2A (r=-0.943, p=0.005) were found.”.
- line 434: although you have shown the correlation with some echocardiographic parameters, no differences in the gene expression was shown between the groups (especially between DMVD and DMVD-PH groups) therefore the conclusion seems precipitative
- From the suggestion, we removed the mentioned context that concluded precipitately and added some comments on working with larger sample to clarify this point. (The corrected context is shown in the next comment.)
- lines 432-444: the results of correlation between echocardiographic parameters and gene and protein expression are inconsistent and therefore the conclusions need more discussion to be drawn; tph1 and TPH-1 showed positive correlation with LA/Ao, %FS and PAP - while increase in LA/Ao and PAP is combined with worsening of the cardiac function, the increase in %FS is not unambiguous - the fractional shortening first increases to decrease in end-stage disease; similarly the pERK1/2 and ERK1/2 expression showed positive correlation with LVPWsN, LA and LA/Ao and as LA and LA/Ao increase is correlated with disease progression, the thickening of LV wall is not explicit.
- We added more references to explain the increase in %FS and LV wall along with the disease progression as follows.
“The present study has shown that the relative expression of some targeted genes and proteins in lung, PA, pneumocytes and PAM was strongly correlated with some echocardiographic parameters. The positive correlation between the 5-HT-effector expression in all targeted tissues and cells and LA/Ao, LVIDdN and PAP may support that increase in activation of 5-HT signaling pathway can induce the more echocardiographic changes in dogs with DMVD and PH secondary to DMVD. The studies in animal models with induced PH mentioned that inhibition ofSERT and 5-HT receptors can reduce the cardiac structural changes of the and PAP [69,70,72]. Positive correlation between TPH-1 and %FS, pERK1/2 and LVPWsN were found in the pneumocytes, and between ERK1/2 and LVPWsN was found in PAM. Theses evidence reflected increased %FS and LV wall thickening during the disease progression. The progressive mitral regurgitation in DMVD increases the cardiac workload resulting in LV remodeling [34]. Furthermore, all dogs in DMVD and DMVD+PH groups were treated with pimobendan which was reported to induce the ventricular hypercontractility resulting in greater %FS [73]. Being negatively correlated with PAP of PA tissue and IVSdN of pneumocytes, the relative 5-HTR2A expression was contrasted with the results. Working with the larger sample size may clarify this distinction and prove that the correlation between the echocardiographic parameters and the relative expression of the targeted genes and proteins may lead to the prediction of the changes in the gene and protein expression.”

Round 2
Reviewer 2 Report
Dear Authors,
Thank you for all the corrections you have made.
I have no further comments.
Author Response
Thank you very much.